# Optimizing bike-sharing station locations: A machine learning and artificial neural networks approach using geospatial and demographic data

Marek Weis[ID][☼][*], Wojciech Dawid[☼]

Faculty of Civil Engineering and Geodesy, Military University of Technology in Warsaw, Warsaw, Poland

☼ These authors contributed equally to this work.
* marek.weis@wat.edu.pl

## Abstract

In the modern world, public transportation amenities are noticeably on the rise, with urban bike-sharing systems becoming well-established in many major cities. However, not all cities have these systems, and planning optimal locations for bike-sharing stations is a complex task that requires consideration of many factors. To address this, the authors of this research paper developed a model to predict suitable locations for bike-sharing stations, utilizing machine learning techniques and artificial neural networks. These techniques utilized land cover and demographic data to train the model, achieving a high accuracy of 0.977. The predicted bike-sharing stations not only align with existing networks but also support their expansion, as many suggested locations are near major intersections and public transportation stops, confirming their suitability for the urban bike network. Additionally, the model was applied to Rzeszów, a city without a current bike-sharing system, where it successfully identified optimal locations for new stations. This demonstrates the methodology's practical applicability and its valuable support for planning bike-sharing infrastructure in urban areas.

## Introduction

In recent years, urban bike-sharing systems have experienced significant development and innovation, reshaping urban transportation. Technological advancements, growing environmental awareness, and the demand for convenient, sustainable city mobility are driving this progress. Modern bike-sharing programs now feature user-friendly mobile apps, GPS tracking, and electric bikes, enhancing accessibility and user comfort. Cities worldwide have adopted these systems to ease traffic congestion, reduce carbon emissions, and promote healthier lifestyles. The integration of bike-sharing with public transportation networks has further solidified their role in the urban mobility ecosystem. Consequently, bike-sharing has evolved from a niche service into a crucial element of smart city infrastructure, aligning with broader trends

**Data availability statement:** All work files are available from the figshare database (DOI URL: https://doi.org/10.6084/m9.figshare.31566433).

**Funding:** A publication fee is funded by Military University of Technology, Faculty of Civil Engineering and Geodesy, Grant Number 531-000130-W400-22. The funders had no role in study design, data collection and analysis, decision to publish, or preparation of the manuscript.

**Competing interests:** The authors have declared that no competing interests exist.

toward sustainable development. The COVID-19 pandemic accelerated the adoption of bike-sharing as people sought safer, socially distanced transport options [1], prompting cities to expand bike lanes and invest in cycling-supportive infrastructure, ensuring the continued growth and integration of bike-sharing in urban planning.

In many cities around the world, urban bike-sharing systems are already successfully operating. A crucial aspect of their implementation is the optimal placement of bike rental stations. This issue is typically addressed by municipal authorities in collaboration with bike-sharing system operators. Companies specializing in managing bike-sharing systems, such as Lime [2], Jump [3], and Nextbike [4], are also involved in this process. Municipal authorities conduct analyses to determine optimal station locations, taking into account factors such as population density, the availability of cycling infrastructure, proximity to tourist attractions, business centres, and transportation hubs [5].

### Related works

Many researchers have focused on bike-sharing topics in their studies, with a significant portion examining patterns of urban bike-sharing usage. For instance, in the publication by Yang et al. [6], scientists investigated how the introduction of a new metro line affected the dynamics of bike-sharing stations in Nanchang, China, using spatial statistics and graph-based approaches to analyse changes in traveller behaviour. In another study by Yao and Feng [7], the authors proposed the TS-SBFP (Two-Stream Station-level Bike-sharing Flow Prediction) framework to accurately predict bike-sharing flows at the station level. Experiments conducted with data from Chicago, the District of Columbia, and New York (USA) confirmed the framework's effectiveness in predicting bike flows. Despite the global adoption of bike-sharing systems, it is noteworthy that research on identifying patterns of urban bike-sharing movement and predicting demand is primarily conducted in China and the USA. This includes studies by J. Chen et al. [8] (where the Bicycle Station Dynamic Planning (BSDP) on basis of four modules: bicycle drop-off location clustering, bicycle-station graph modelling, bicycle-station location prediction, and bicycle-station layout recommendation is created with the use of Gated Graph Neural Network), Lin et al. [9] (where the spatiotemporal patterns of dockless bike sharing demand as well as factors influencing these patterns were examined on basis of bicycle trip data from Mobike, Point of Interest (POI) data and smart card data) and Zhu et al. [10] (where comprehensive framework for analysing BIPTS (Bike-sharing for Integrated Public Transport Systems) commuting demands, is developed)in Beijing, as well as Gong et al. [11] (where machine learning and multiscale geographically weighted regression models at both station and neighbourhood scales for a comprehensive analysis of the relationships between the effects of natural environment and street visual quality on cycling in the spatial dimension were examined), Harikrishnakumar and Nannapaneni [12] (where quantum computing algorithms were implemented to provide computational speedup in comparison with classical algorithms as a solution framework for implementing Quantum Bayesian Networks for bike demand prediction during weekdays and during weekends), and Liang et al. [13] (where the authors focus on the trip

generation problem for bike-sharing systems expansion, and propose a graph neural network approach to predicting the station-level demand based on multi-source urban built environment data) in New York City.

Another significant area of research focuses on determining optimal bike station locations. In the study by Nikiforiadis et al. [14], a methodological approach was developed to optimize bike station locations to maximize demand and area coverage while minimizing the need for bike redistribution. This approach treats station selection as a multi-criteria optimization problem, demonstrating that different weights assigned to goals influence station distribution. For instance, prioritizing demand coverage favours waterfront locations, while area coverage favours central city locations. Similarly, in the article by Banerjee et al. [15], the authors conducted spatial analyses to identify three new bike station locations for a bike-sharing program in Baltimore, USA. By analysing over 1.6 million GPS coordinates, they developed a methodology based on a modified Huff gravity model to determine optimal bike station locations, considering factors such as proximity to public transportation, attractions, and restaurants. Additionally, many researchers have focused on the optimal placement of bike stations on university campuses, employing spatial analyses ([16] – the spatial analysis based on data about street lines, cycling lines and numbers of inhabitants and others with the use of Extension Network Analyst for ArcGIS; [17] – GIS-based method to calculate the spatial distribution of the potential demand for trips, locate stations using location–allocation models, determine station capacity and define the characteristics of the demand for stations; [18] – an origin–destination (O-D) matrix to identify appropriate bike station locations at the Morgan State University campus) and various mathematical models [19], where a case study is applied to the Gaziantep University campus in order to find possible locations of the stations for users (students) and the mathematical models of P-center and P-median are used to build possible stations and to allocate demand points to the opened stations.

Given the complexity of analysing urban bike-sharing usage patterns, which involves non-linear relationships between variables such as weather conditions, time of day, and days of the week, many researchers turn to computational intelligence algorithms. For example, the study by Zi et al. [20] introduces the TAGCN (Temporal Attention Graph Convolutional Network) model, which predicts bike check-out and check-in numbers at each station. This model effectively captures spatial and temporal dependencies across various time granularities, significantly enhancing station-level demand prediction and rebalancing in bike-sharing systems, as demonstrated with four real-world datasets from the Divvy Bike System in Chicago. Meanwhile, Chen et al. [21] compare five different architectures based on recurrent neural networks (RNNs) for predicting real-time demand for bike rentals and returns at individual stations. Additionally, Boufidis et al. [22] developed a tool that uses regression algorithms to predict and visualize bike rental and return demand at each station. These studies underscore the increasing reliance on advanced computational techniques to manage the complexities of bike-sharing systems effectively.

## Research purpose

Based on the reviewed literature, it is clear that the success of bike-sharing programs heavily depends on the strategic placement of bike stations. While many cities have implemented urban bike-sharing systems, numerous others have yet to effectively introduce them. This research aims to develop a model that leverages spatial data (such as topographical information, distances from public transportation stations, and proximity to universities) and demographic data to automatically determine optimal bike station locations. This study addresses the following research questions:

- Can a universal model be developed using spatial and demographic data to identify optimal locations for bike-sharing stations?

- Which variables have the most significant impact on selecting these optimal locations?

- Which machine learning method or neural network architecture yields the most accurate predictions for bike station placement?

Answering these questions will contribute to the existing body of literature, as previous studies have not extensively explored the use of computational intelligence algorithms for determining optimal bike station locations based on spatial and demographic datasets. Moreover, the results of this research hold substantial practical value, potentially optimizing the design and implementation of bike-sharing systems.

## Materials and methods

### Permits for the work

During the process of collecting and processing the data for the work, no permits were required as only public data with open access were used.

### Study area and data used

The research focused on two Polish cities with existing bike-sharing systems, including a network of bike stations. The primary area of study was Warsaw, while the developed method was tested in Łódź. This was performed in order to check weather the proposed method is appropriate to use regardless of the location, being more precise – regardless of the characteristics of the city.

Warsaw, the capital of Poland, is situated in the central-eastern part of the country, with coordinates at 53°13′56″N and 21°00′30″E. Covering an area of 517.2 km², it is home to nearly 2 million residents, making it the largest city in Poland [23]. The city features a lowland terrain with elevation ranging from 78 to 121 meters above sea level and is known for its numerous tourist and business attractions, well-developed road network, cycling infrastructure, and advanced urban bike-sharing system.

Łódź, the fourth largest city in Poland, spans 293.3 km² and has a population of up to 700,000 [24]. Located in the central part of the country, its central coordinates are 51°46′36″N and 19°27′17″E. The city's elevation ranges from 162 to 278.5 meters above sea level. Although it has fewer bike paths compared to Warsaw, Łódź boasts a well-developed transportation infrastructure and, like Warsaw, features an extensive network of urban bike-sharing stations. Fig 1 illustrates the locations of the research and testing areas on a map of Poland.

The research utilized several data sources, with the primary one being the Topographic Objects Database (BDOT10k). This open vector dataset, provided by the Chief Geodesy and Cartography Office (GUGiK), is detailed to a scale of 1:10,000. It includes spatial locations of topographic objects and their basic descriptive characteristics. The thematic scope of BDOT10k encompasses transportation networks, buildings, structures, equipment, and land use complexes, all of which are highly relevant for this study. The database is continuously updated and is available free of charge. It can be downloaded from the National Geoportal [25]. Fig 2a displays the BDOT10k visualization for the Warsaw area.

Population density, a key parameter for assessing the attractiveness of potential bike station locations, was sourced from the Global Human Settlement Population Grid (GHS-POP) provided by the European Commission [26]. This dataset comprises raster files with a resolution of 100 meters, where the values indicate population density at specific locations. The data used in this research are from the year 2023. Fig 2b illustrates the population density visualization for Warsaw based on the GHS-POP data.

The research also incorporated data on the locations of individual bike stations in both Warsaw (the research area) and Łódź (the testing area). For Warsaw, the coordinates were obtained as text files from the Warsaw City Hall portal. Since data for Łódź were not available for direct download, the bike station locations were manually vectorized. Fig 2c and 2d display the distribution of bike stations in vector format for both study areas.

### Methods applied

**Research hypothesis and methodology overview.** The primary goal of the research was to develop a method for determining optimal bike station locations in any city using machine learning (ML) and artificial neural networks (ANN) based on existing stations of bike-sharing stations. From mathematical point of view, the most important question is

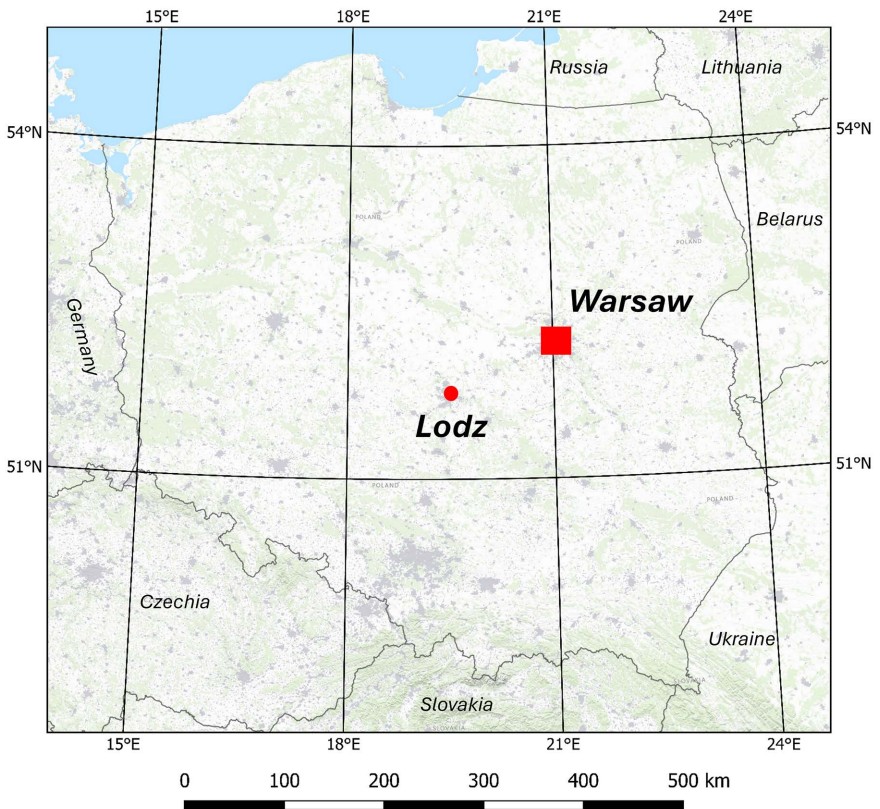

**Fig 1. The locations of Warsaw and Łódź in Poland, WGS-84 coordinate system.** The base map created using USGS National Map Viewer (https://www.usgs.gov/tools/national-map-viewer), shared under the CC BY 4.0 licence (https://creativecommons.org/licenses/by/4.0/deed.en).

whether it is possible to reach sufficient values of metrics of results of predictions made by ANN. Main constraints might be connected with the lack or improper resolution of learning data for ANN (both spatial and demographic) and with inability to use the models created regardless of the location, but it will be discussed more widely in the following sections of the study. The workflow of the methodology is illustrated in Fig 3, with a detailed description of each step provided in the following subsections.

   **Data selection and preprocessing.** The first step involved selecting the input data (see Fig 3.1). This step was crucial because not all collected data were essential for the research's effective execution. It was necessary to determine which data and parameters identified a location as suitable for a bike station and which might be counterproductive. S1 Table lists elements that either support or hinder the suitability of bike station locations, along with the sources of the data used for this analysis. The categorization of land cover elements was influenced by societal transportation preferences, including employment needs, consumption patterns, and cultural and tourism activities, while also taking into account existing infrastructure that facilitates various forms of physical movement. Standardization of types od supportive and non-supportive spatial information is unjustified as the division deeply depends on the set of spatial data used in the study. Different dataset could contain differently described data, which might affect the work of algorithms that would be created on the basis of data divided by aforementioned standardisation. In general, the supportive places refer to communication lanes, places of high density of population (both permanent and temporal) and proximity to often visited places, while non-supportive places are the places where it is impossible or prohibited to build anything (including bike stations). The selected data were then used as input for the subsequent processing steps.

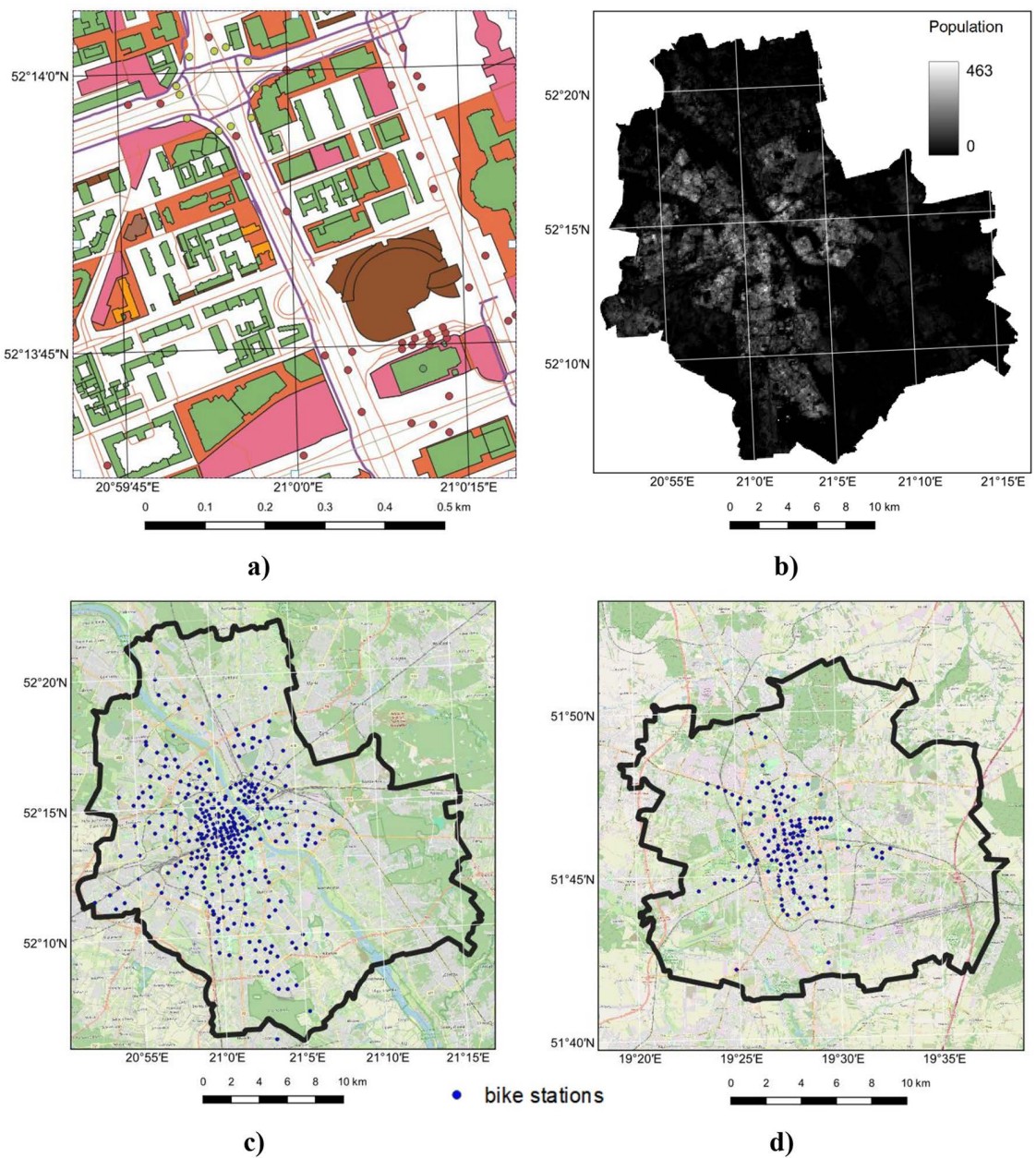

**Fig 2. Visualization of BDOT10k for the Warsaw area (a), population density in Warsaw based on the GHS-POP model (b), distribution of bike stations in Warsaw (c) and Łódź (d) on an OpenStreetMap (OSM) base map, WGS-84 coordinate system.** The base map in panel a created with Quantum Geographic Information System (QGIS) software using Polish National Geoportal (https://www.geoportal.gov.pl/pl/dane/baza-danych-obiek-tow-topograficznych-bdot10k), shared under the CC BY 4.0 licence (https://creativecommons.org/licenses/by/4.0/deed.en) on basis of the permission granted by the owner of the data (Head Office of Geodesy and Cartography). The base map in panel b created with Quantum Geographic Information System (QGIS) software using GHSL data (https://human-settlement.emergency.copernicus.eu/index_op.php), shared under the CC BY 4.0 licence (https://creativecommons.org/licenses/by/4.0/deed.en). The base map in panel c and panel d created with Quantum Geographic Information System (QGIS) software using OpenStreetMap (https://www.openstreetmap.org/), shared under the Open Database Licence (http://www.opendatacommons.org/licenses/odbl).

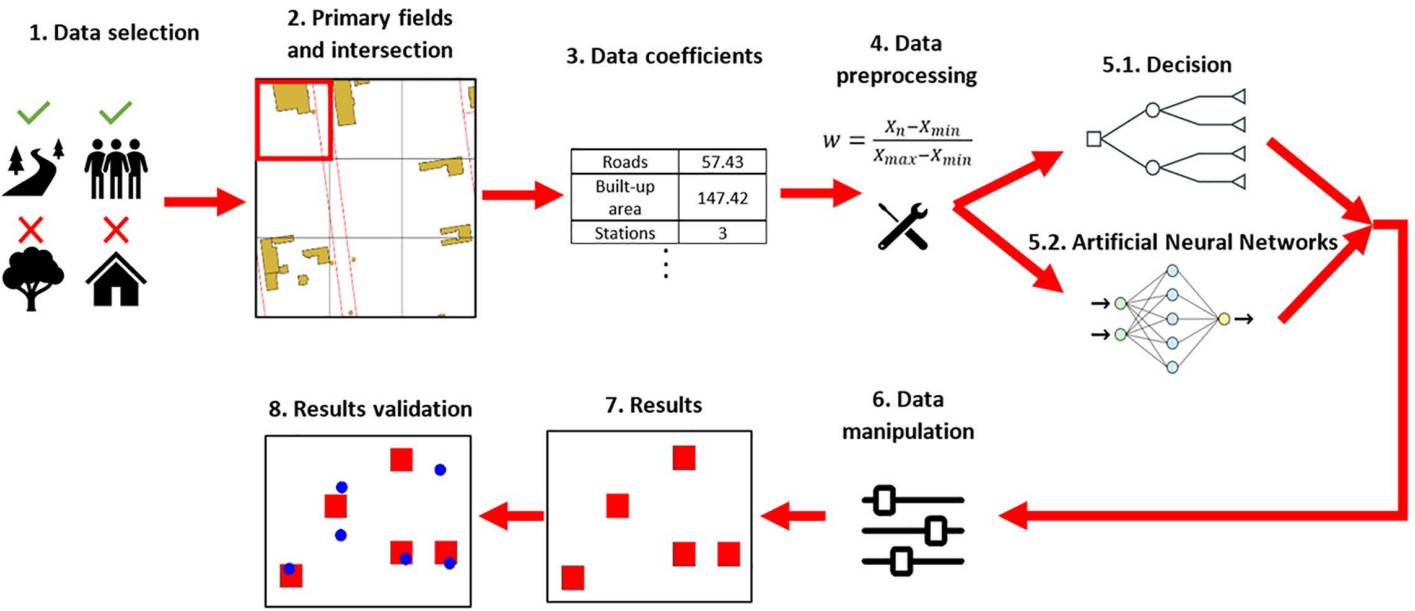

**Fig 3. A diagram illustrating the workflow of the methodology.**

In the next step, a grid of primary fields was created, consisting of squares measuring 100x100 meters (Fig 3.2). This grid served as the fundamental layer for predicting the presence of a bike station within each cell. The size was selected so that the maximum distance between two stations would not exceed 300 meters (rounded from the exact doubled diagonal of the base field, precise value: 282.84 meters), ensuring that walking between them would take less than 5 minutes (assuming a slow walking pace of 3.6 km/h) [27]. Nevertheless, undoubtely the most important reason for choosing grid size of 100x100 meters was a limitation imposed by the available size of GHS-POP population model. The only sizes possible to download and, by extension, to use in the study were 100x100 [m], 1x1 [km], 3 [arcsec] and 30 [arcsec]. Grid size smaller than 100x100 [m] would lead to false information as the partition of cells with the use of resampling techniques does not create or extract any new information and worsens the quality of data. Therefore, as it was unjustified to use any smaller grid than 100x100 [m] (neighbouring primary fields would have the same demographic information although it would not be true), the smallest possible grid size was used. Larger sizes also were not taken into consideration as it is far less precise to indicate such a big area where a small bike station should be placed. Also, the distance between nearest possible locations of stations should not exceed 5 minutes of slow walk, as it was mentioned above. Therefore, a grid size of 100x100 meters was chosen as the most appropriate.

Attributes related to the input data were added to each cell of the primary field grid through the intersection process (Fig 3.3). These attributes included the number of points within the primary field (for point layers), the total length of lines within the field (for linear layers), and the total area of elements in the field (for surface layers). Fig 4 illustrates the overlay of the primary field grid, roads, and existing bike stations within the study area, as well as the results of intersecting these objects on the example of a single primary field.

In the same manner, the distance from the city centre and important landmarks (museums, schools, cultural sites, etc.), as well as the population density in the area, were also associated with the primary grid cells.

The final step in the preliminary data preparation involved calculating the coefficients for each attribute recorded in the grid of primary fields table (Fig 3.4) using Min-Max Normalization [28]. This normalization was essential to unify the units of the input data, ensuring that all attributes were on a comparable scale. Each attribute—whether representing

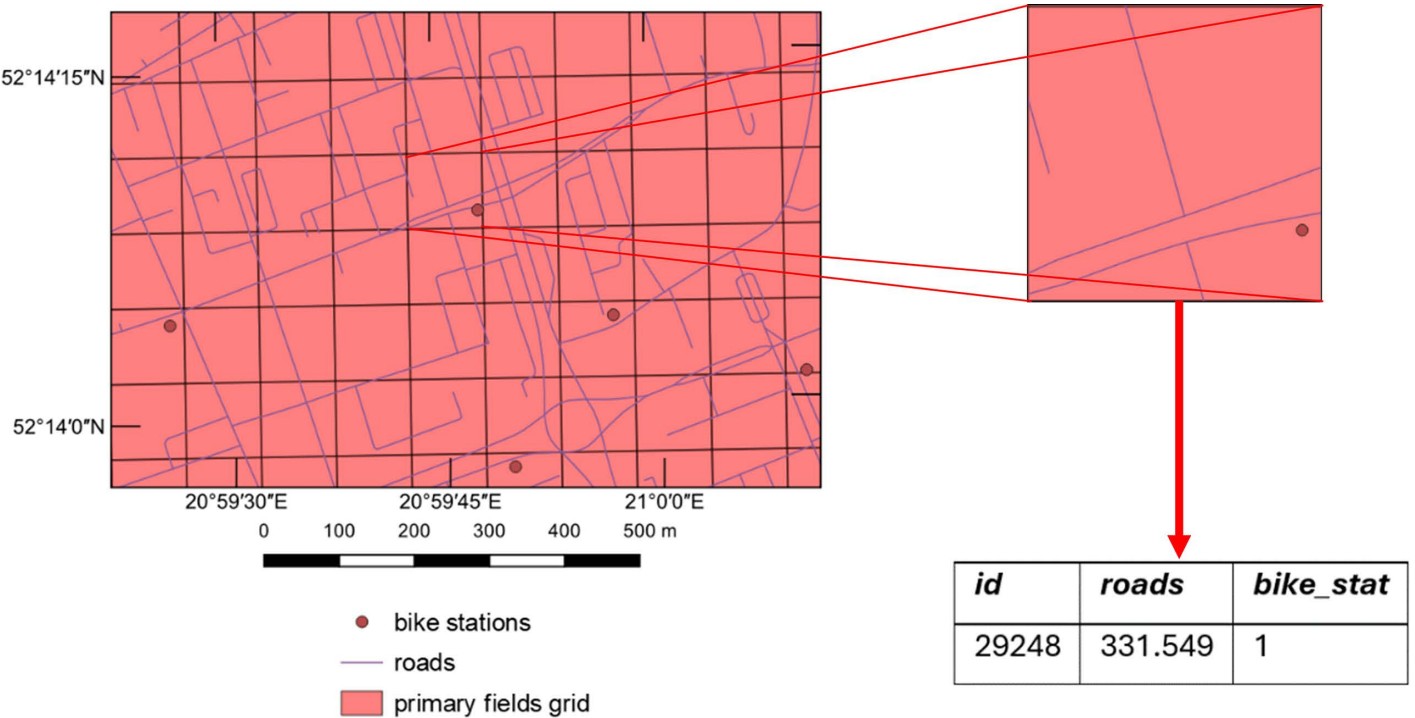

**Fig 4. Overlay of the primary field grid, bike routes, and bike stations in the study area, along with the results of the intersection analysis.**

the number of points, line length, or surface area within a given primary field—was normalized according to the following formula (1):

$$w = \frac{X_n - X_{min}}{X_{max} - X_{min}}$$

(1)

where:

 $w$ – parameter coefficient value,

 $X_n$ – particular coefficients value,

 $X_{min}$ – minimum value of the parameter for the entire dataset,

 $X_{max}$ – maximum value of the parameter for the entire dataset.

 This step completed the preliminary data processing and prepared the data as input for machine learning and artificial neural networks.

 **Training a model.** The data prepared in the previous step were then utilized in the next stage, which focused on applying computational intelligence algorithms to determine the optimal locations for bike-sharing stations. This was accomplished using two main prediction methods: decision tree learning (including its associated random forest method) and artificial neural networks (ANNs), as illustrated in Fig 3.5. The decision tree method uses a tree structure to represent hypotheses, where nodes test input data, branches show test outcomes, and leaves indicate decisions. The random forest method, an extension of decision trees, builds multiple trees during training and aggregates their outputs for predictions [29]. Artificial neural networks (ANNs) mimic neuron functions in the brain, processing data through a sequence of neurons using activation functions, with the network's performance evaluated by a cost function [20]. In both cases, the

data processing and prediction tasks were executed using the Python programming language and its specialized libraries for handling tabular data, such as Pandas and GeoPandas.

It should be articulated that although the use of machine learning and ANNs techniques in predicting best possible locations of particular types of spatial objects has been widely discussed in numerous works (mentioned in "Related works" section), it appears that the proposed methodology in connection with the use of spatial vector data and raster demographic data has not been deeply examined yet. Previous works discuss convolutional neural networks, graph neural networks [20], or recurrent neural networks [21] and the use of other data than it this study. Therefore, although it is similar to some existing works, the proposed methodology introduces innovation relative to existing studies.

A crucial step in data preprocessing was resampling, needed to address imbalanced training data. Since the methodology used primary fields, only a small portion of these fields within the city contained at least one bike-sharing station, creating significant discrepancies in the input data that could hinder the learning process. An introduction of an imbalance-handling technique is important because the AI methods of making prediction models perform poor results quality when the data is imbalanced, which is proved in [30]. To mitigate this, over-sampling was applied to primary fields with bike-sharing stations using the "RandomOverSampler" method from "imblearn" library in Python. It was implemented to make the data classified as 0 (no bike station in primary field) and 1 (bike station in primary field) equal. This tool generates new samples by randomly duplicating the minority class and adjusting the majority class samples until balance is achieved, ensuring equal representation of both classes [31]. Important advantage of this method (often seen as disadvantage, though) is fact that it does not create any new information, but only reproduces existing. Thanks to that, the risk of making wrong predictions a priori (while performing preprocessing of data) is reduced as the algorithm does not make any variations nor interfere in original data, which can trigger an occurrence of the data unlike to the rest. Therefore, as there were much less samples containing bike stations than samples without them, in order to handle possible low specificity of the prediction results, imbalance-handling technique described above was applied.

In the first part of the study, predictions were made using the decision tree method, followed by the random forest method. Both methods classified the primary fields into two classes: 0 (no bike-sharing station present) and 1 (bike-sharing station present). Fig 5 illustrates the visualization of a sample decision tree used in the research.

In order to introduce and examine other methods, prediction was performed using ANN. The pre-processed tabular data was used to train the network, with parameters set manually. These parameters included:

- Activation functions: RELU, Sigmoid, and Tanh were considered.

- Number of hidden layers: Ranging from 2 to 4.

- Number of hidden neurons: Selected based on learning accuracy (Fig 6).

- Loss function: Mean Square Error (MSE) and Cross-Entropy Loss Function were considered.

- Number of epochs: Ranged from 50 to 400.

These parameters were adjusted to achieve the best possible results (Fig 3.6). Validation data constituted 20% of all data used for training.

Fig 6 shows the accuracy of the neural network training for different numbers of hidden neurons, depending on the number of hidden layers. Fig 6a refers to a network structure with one hidden layer (16-x-1), while Fig 6b) refers to a network with two hidden layers (16-x-y-1). In the latter case, the plot presents the combination with the best obtained accuracy parameters for each network with a given neuron count "x" (ranging from 3 to 15). Accuracy was measured for three datasets: training, validation, and test. Based on this data, the optimal neural network structure was selected (Fig 7).

In the research on the optimal neural network structure, different activation functions were also tested: RELU, Sigmoid, and Tanh [32]. Modifications were made for two cases: classification and linear regression. For the regression results

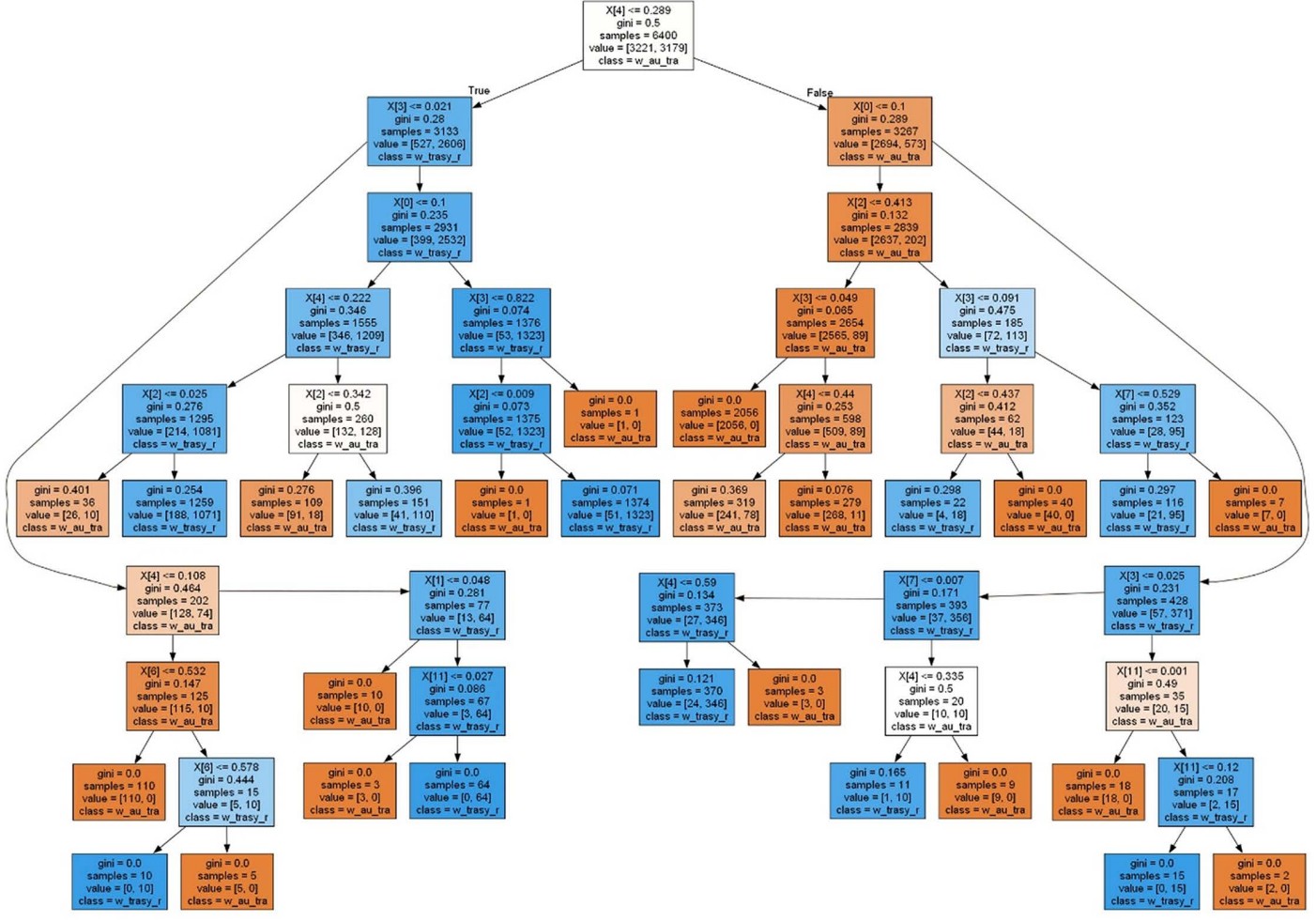

**Fig 5. Diagram visualizing the decision tree used in the research.**

(which have a continuous distribution), an optimal cutoff threshold (0.99) was determined. Values below this threshold were converted to 0, and values above were converted to 1. This allowed for the classification of linear regression results to clearly determine whether a given primary field is conducive or not to the establishment of a bike station. Accuracy was used as the criterion for evaluating the performance of the neural network. The tests were concluded when no modification of any network parameter resulted in a higher accuracy value than the previous one (Fig 3.7).

Fig 7 shows the structure of the neural network model that exhibited the best accuracy parameters during the search for the optimal model. Based on Fig 6, it can be concluded that the neural network structure with two hidden layers yields better results in terms of learning accuracy. Therefore, this network structure was used for further research. Additionally, fewer neurons in these layers may lead to underfitting, while more neurons could cause overfitting [33]. This led to the selection of a number of neurons from the middle range. Within this range, the structure 16-8-4-1 displayed the best accuracy parameters as well as the highest learning stability, so this structure was used in the subsequent stages of the research.

**Results testing.** The next step involved validating the predictive models on an independent test area, which was Łódź (Fig 3.8). To do this, data with the same structure as that of the research area (Warsaw) was collected and processed

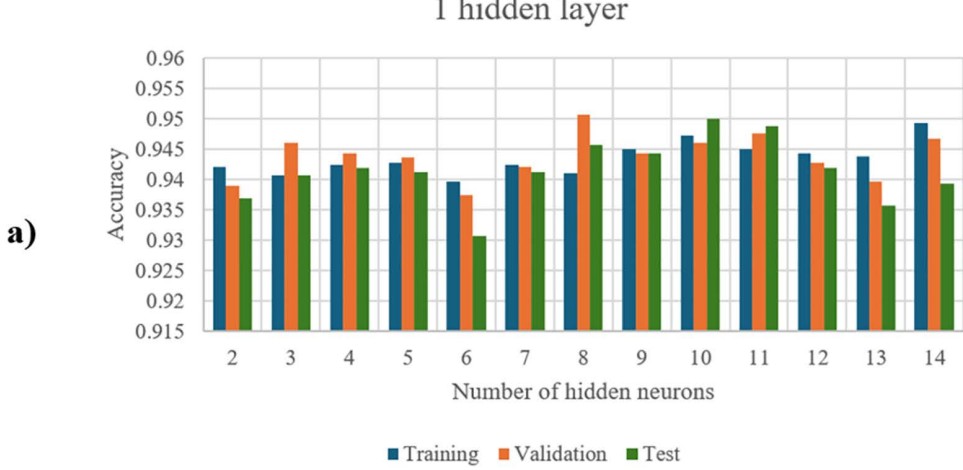

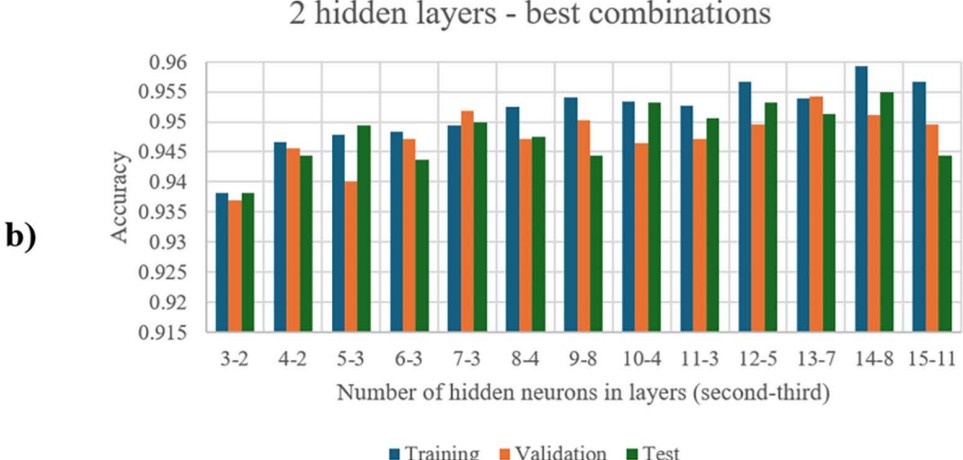

**Fig 6. Dependency of neural network training accuracy on the number of hidden neurons for combinations of one (a) or two (b) hidden layers.**

similarly, allowing for the application of the developed models. The results of this validation are presented in the following section.

## Results

Following the previously described methodology, neural network training and prediction using a decision tree were carried out. As a result of the tests, for each combination of applied methods, the best result was obtained, which was the outcome of selecting the input parameters. Only in the case of the ANN, it was decided to present two best results, which outperformed the other methods in terms of quality. The validation results were presented as a confusion matrix, where 0−0 indicates the correct classification of a field without a bike station, 1−1 indicates the correct classification of a field with a station, while the 0−1 and 1−0 configurations represent incorrect classifications. The results include the following cases:

• Case 1: Decision tree method with result classification (S2 Table);

• Case 2: Decision tree method with linear regression of results (S3 Table);

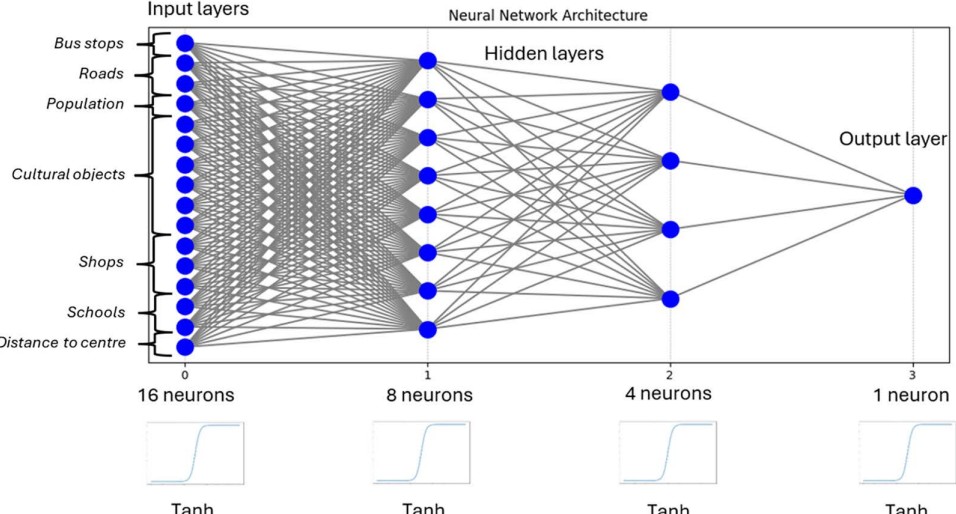

**Fig 7. The neural network structure with the best accuracy parameters among those tested in the research: 2 hidden layers and 1 activation function.**

- Case 3: Artificial neural network method with result classification (Fig 8a and S4 Table);

- Case 4: Artificial neural network method with linear regression of results and the use of the RELU activation function for the first 3 layers and Sigmoid for the output layer, over-sampling technique implemented (Fig 8b and S5 Table);

- Case 5: Artificial neural network method with linear regression of results and the use of the tanh activation function for all layers, over-sampling technique implemented (Fig 8c and S6 Table).

Additionally, for each method, the accuracy of the model generated using it (whether a decision tree or a neural network) was determined, along with its precision, sensitivity, and specificity [34]. This is presented in S7 Table.

The table above clearly shows that the model used in Case 5 – ANN with linear regression of results and the tanh activation function – achieved the best accuracy. It also performed the lowest difference between the values of sensitivity and specificity (although it is still visibly high, the use of over-sampling technique lowers it, preventing the model from even more significant differences that are the proof of class imbalance). Based on these findings, this model was selected for further research activities. All results were converted into vector file format and visualized using GIS software (Fig 9). For classification methods, the fields were divided in a binary manner, while for linear regression, the best empirical cutoff threshold (0.99) was applied to highlight the fields with the optimal locations for bike stations.

A prediction of the bike station network distribution for Warsaw was also performed using the model with the highest accuracy. The prediction accuracy for Warsaw's bike stations was approximately 0.981, and the results are shown in Fig 10.

## Discussion

The obtained results lead to several important conclusions, both regarding the prediction process for urban bike station locations and the comparison of these predictions with the actual distribution of stations in the test area.

### Influence of spatial factors

The decision tree method enabled the assessment of how individual factors, derived from basic field characteristics, affect the suitability of locations for urban bike stations and influence the prediction results for their distribution. The correlation

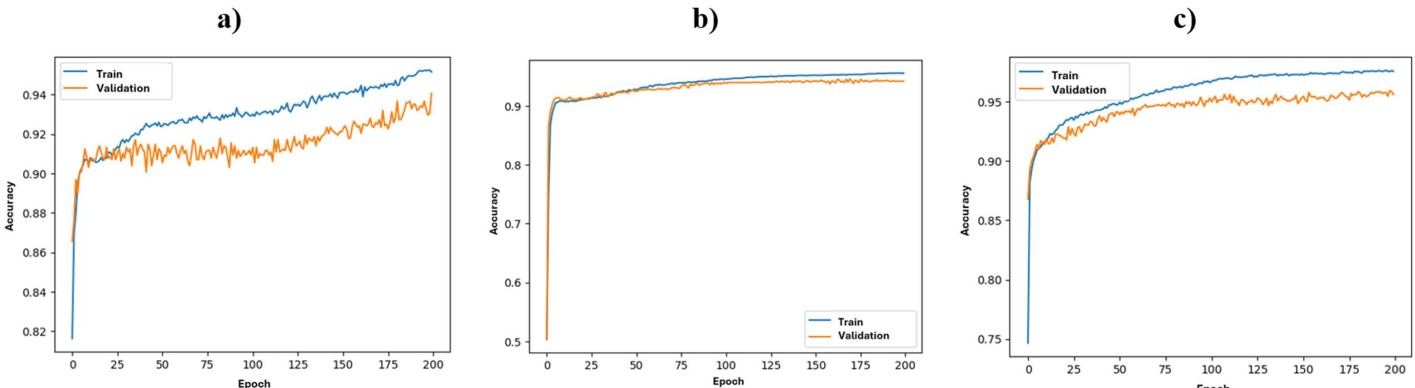

**Fig 8. Learning curves for neural networks for cases: 3 (a), 4 (b), and 5 (c).**

between the likelihood of bike station locations and the coefficients of specific parameters – both supporting and not supporting bike station placement – was analysed. The results are presented in S8 Table.

As shown in the table above, the distance from the city centre has the greatest impact on the placement of urban bike stations. The city centre, with its high concentration of tourist attractions and public amenities, benefits from bike stations that facilitate easy access to these destinations. Proximity to public transportation stops is also crucial, as positioning bike stations near these stops reduces travel time to subsequent transfers or to areas not covered by public transit. The third most significant factor is the proximity to roads where city bikes can be used. Population density ranks fourth, as placing bike stations in densely populated areas ensures that the maximum number of potential users can benefit from them. This is particularly relevant for children traveling to school or students to universities, which is why proximity to schools and educational institutions is the fifth factor. Interestingly, bike paths do not rank among the top factors. This is probably due to the current limitations of the bike path network; not every bike station is located near a dedicated bike path, which affects the algorithm's emphasis on their importance. Moreover, bike paths network is underdeveloped as there are places of discontinuity between lanes. It introduces the situation in which there are many bike stations both in the vicinity and further away from bike paths. Therefore, the algorithm cannot undoubtedly state if the bicycle lane is a favourable factor for bike station placement or not. In a more extensive network, bike paths would likely play a more significant role in determining station locations.

## Comparison of results

With the collected data on the locations of actual bike stations from urban bike systems operating in Warsaw and Łódź, it is possible to validate the obtained results by comparing the predicted bike station locations with their actual positions. For this purpose, the percentage of predicted stations within 100 meters, 200 meters, and 300 meters of the nearest existing station was determined. The results are presented in S9 Table. Similar values of the given percentage statistics are a proof that the proposed method is suitable to use with the data for any city.

For both cities, the average distance between existing and proposed bike station locations was calculated using nearest neighbour analysis. This distance is approximately 231 meters for Warsaw (training area) and about 314 meters for Łódź (test area). These values include all stations, including those in suburban areas where distances are naturally larger. A closer examination reveals that some locations overlap completely, while others are situated in areas with similar characteristics, such as major intersections or public transportation stops. In both cities, the highest density of bike stations is found in the city centre, with density decreasing as the distance from the centre grows, as illustrated in Figs 9e and 10.

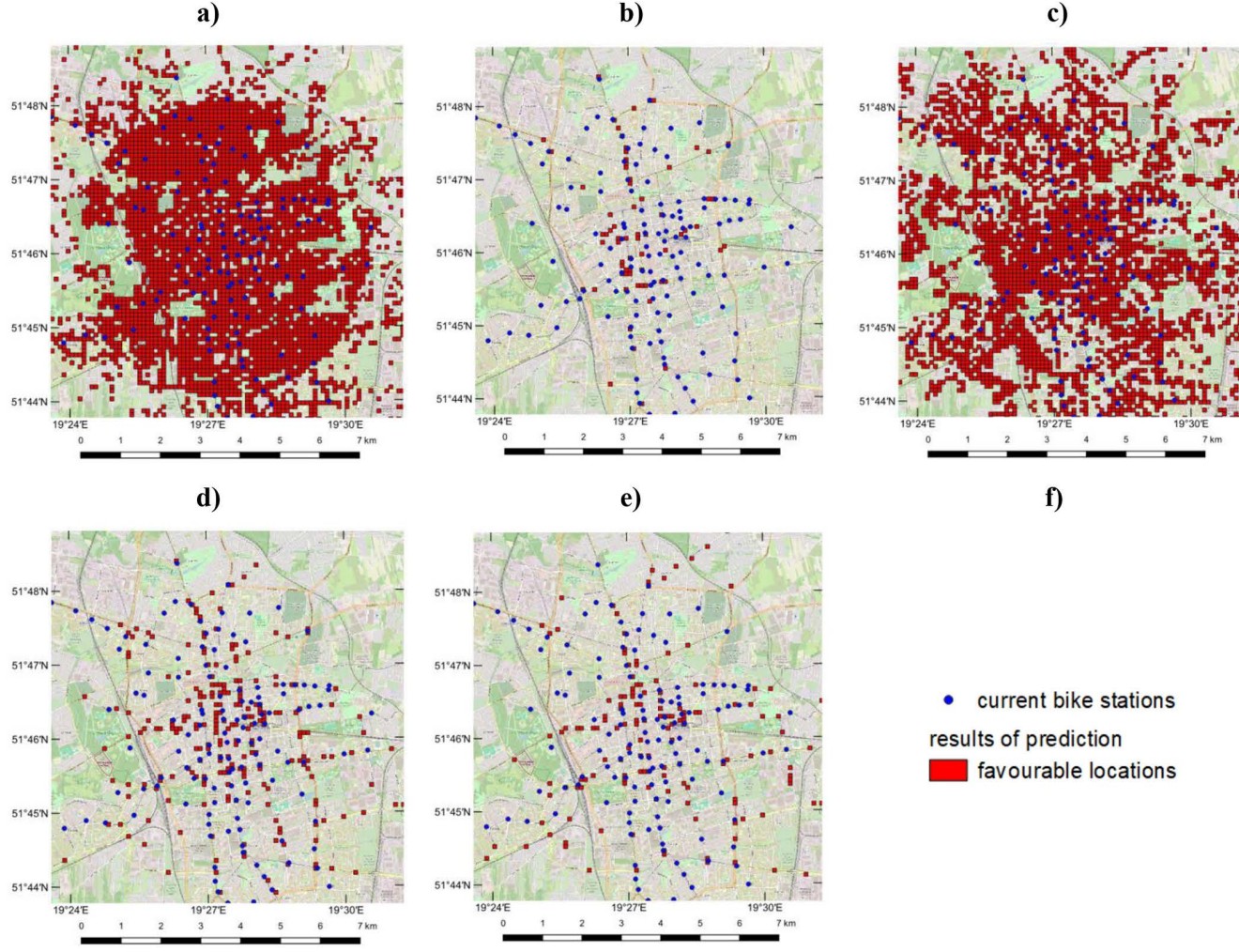

**Fig 9. Visualization of prediction results for cases: (a) – 1, (b) – 2, (c) – 3, (d) – 4, (e) – 5.** All base maps created with Quantum Geographic Information System (QGIS) software using OpenStreetMap (https://www.openstreetmap.org/), shared under the Open Database Licence (http://www.opendata-commons.org/licenses/odbl).

Additionally, in many cases, the actual location of a bike station matched the primary fields identified as optimal for new bike stations, as shown in Fig 11.

There were also instances where the suggested locations for bike stations were correctly identified, even though no actual station was present at those sites. Analysis of the characteristics of the surroundings of the actual bike station locations reveals that stations are generally situated at intersections, along major roads, and in the city centre, as previously determined. The prediction results for bike station locations exhibit the same pattern in relation to their surroundings for most cases – suggesting locations at intersections and along major roads, with the highest density in the central part of the city. It is Falso worth noting that all convenient locations for bike station placement are identified, which sometimes means pointing out several adjacent primary fields. This requires selecting one of them for the best possible bike station location, but the results of the automatic prediction allow the decision-maker to focus only on a significantly narrowed area. This is illustrated in Fig 12.

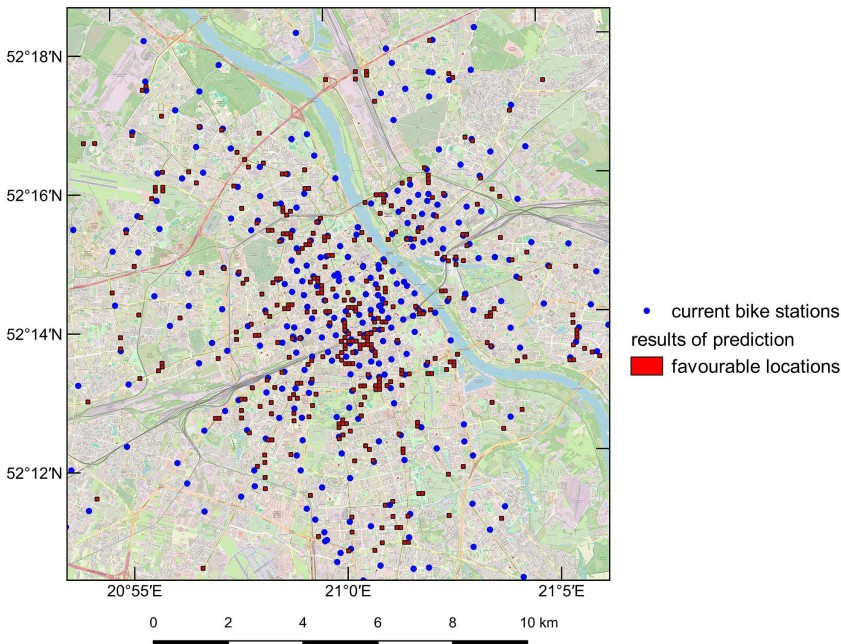

**Fig 10. Prediction results for the bike station network locations in Warsaw using ANN with linear regression and the tanh activation function.** The base map created with Quantum Geographic Information System (QGIS) software using OpenStreetMap (https://www.openstreetmap.org/), shared under the Open Database Licence (http://www.opendatacommons.org/licenses/odbl).

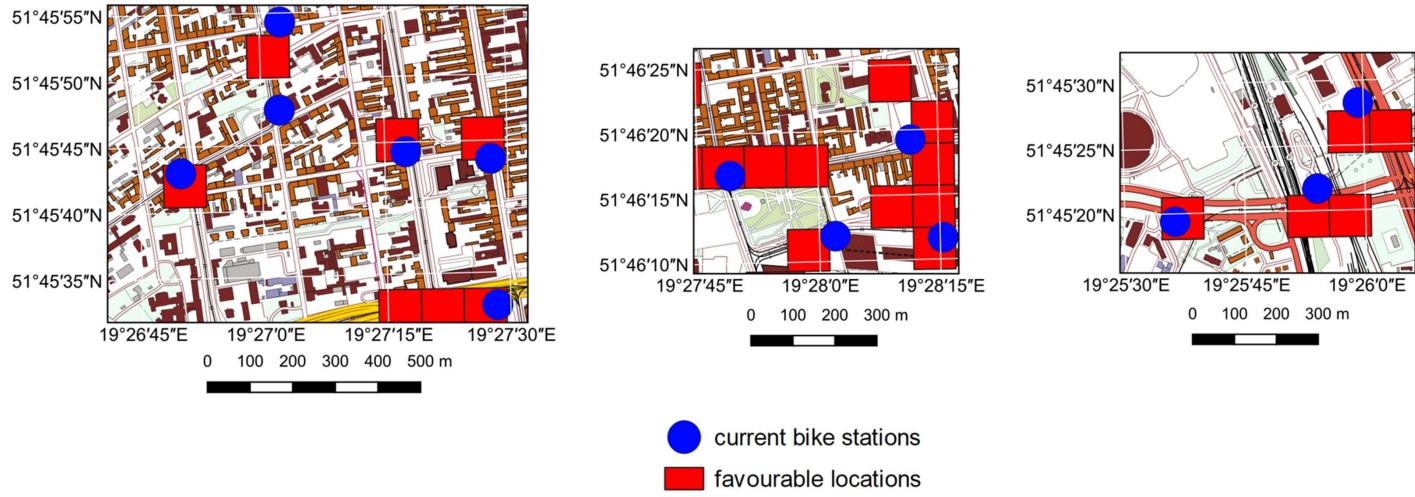

**Fig 11. Locations identified in the research as suitable for bike stations in relation to their actual placement in Łódź.** All base maps created with Quantum Geographic Information System (QGIS) software using Polish National Geoportal (https://www.geoportal.gov.pl/pl/dane/baza-danych-obiek-tow-topograficznych-bdot10k), shared under the CC BY 4.0 licence (https://creativecommons.org/licenses/by/4.0/deed.en) on basis of the permission granted by the owner of the data (Head Office of Geodesy and Cartography).

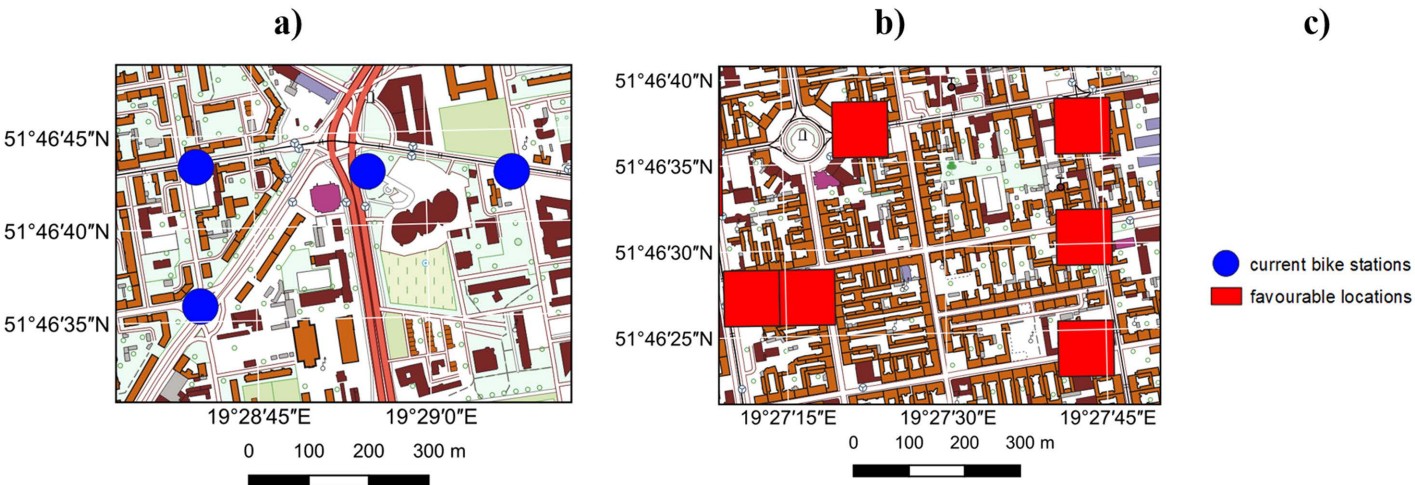

**Fig 12. Bike station locations relative to their surroundings: a) actual, b) predicted.** Both cases show that stations are located at intersections and within the city centre. Both base maps created with Quantum Geographic Information System (QGIS) software using Polish National Geoportal (https://www.geoportal.gov.pl/pl/dane/baza-danych-obiektow-topograficznych-bdot10k), shared under the CC BY 4.0 licence (https://creativecommons.org/licenses/by/4.0/deed.en) on basis of the permission granted by the owner of the data (Head Office of Geodesy and Cartography).

## Spatial analysis of bike station distribution

To determine the spatial characteristics of existing stations and the returned results, a Moran's I analysis was conducted [35]. This involves calculating three principal values: Global Moran's Index (GMI, which measures spatial autocorrelation based on feature locations and attribute values), p-value (the probability that the observed spatial pattern was created by some random process), and z-score (standard deviation). S10 Table provides a summary of the aforementioned values for all the cities where the bike station network distribution predictions were made.

Based on the data in the table above, it can be concluded according to Moran's statistics that the actual location of the stations is characterized by random distribution, while the prediction made by the developed model results in a slight clustering of bike stations.

## Cross-validation of results

There were also cross-validation of obtained results that was performed by the authors. This type of validation refers to multiple division of data between training and test sets in order to assess the effectiveness of machine learning results based on aforementioned data, which is precisely described in [36]. The authors created a script that computes metrics of spatial cross validation and uses information of bike station locations predicted by ANN and information about actual locations of bike stations as input data. The results of its work are presented in S11 Table.

In both cases (for training area as Warsaw and test area as Łódź) we can notice a high value of accuracy that reaches a level of 0.9. It means that the model applied has high ability of classification of fields in favourable and not favourable locations for bike stations. AUC factor (Area Under the ROC Curve), which provides information about the ability of recognizing classes, differs for training and test area. The closer the value is to the value of 1 the better the aforementioned ability, so there are better results for Warsaw than for Łódź. There is also similar situation with MSE (Mean Square Error), which is lower for training area, but it is worth mentioning that for test area is also relatively low. All these values of factors do not emphasize a good quality of the model, but it has to be said that it bases on real locations of bike stations and as they are set up by people, they are not the only proper places for stations and the evaluation of the model by cross-validation is not definitive in that case.

## Formal objective function of results

Another statistics that were taken into account was formal objective function of the results of prediction of bike stations location. It can be defined as a mathematical description of elements to be optimised in the given process, e.g., how advanced the optimisation of the results of learning ANN is ([37]). In this particular case, the objective function was created by adding parts that are responsible for distance and coverage as follows:

$$L_{total} = \alpha \cdot L_{dist} + \beta \cdot L_{cov} \tag{2}$$

where:

- $L_{dist}$ – distance factor;

- $L_{cov}$ – coverage factor;

- $\alpha, \beta$ — weights ($\alpha$ = 0.6, $\beta$ = 0.4; values are given based on importance of each factor).

Factors mentioned in (2) are also defined by formulas given below:

1. Distance factor - measures the precision of spatial prediction — the closer the predicted stations are to the actual ones, the smaller the cost.

$$L_{dist} = \left(\frac{1}{N}\right) \cdot \sum_{i=1}^{N} \frac{(d_i^2)}{1000^2} \tag{3}$$

where:

- $d_i$ – distance (in meters) between the predicted station $i$ and the nearest actual station;

- N – number of predicted stations.

2. Coverage factor – evaluates network coverage — it measures the proportion of actual stations that fall within the coverage radius of any predicted station.

$$L_{cov} = 1 - \left(\frac{1}{M}\right) \cdot \sum_{j=1}^{M} \mathbb{1}\left\{\min_{i} d_{\{ij\}} \leq R\right\} \tag{4}$$

where:

- $M$ — number of actual stations;

- $d_{ij}$ — distance between predicted station $i$ and actual station $j$;

- $R$ — coverage radius;

- $\mathbf{1}\{\cdot\}$ — indicator function (equals 1 if the condition is true, 0 otherwise).

Aforementioned functions were applied to the results by a script written by authors in Python programming language. Several values of R (coverage radius) were taken into consideration (300 [m], 500 [m] and 700 [m]). These values were chosen on basis of aforementioned walking distance ([27]) between stations (300 [m]) and considering two larger values as enlarging distance to one station can lower distance to another. It is shown in S12 Table below. In this table

MAE measures how many meters, on average, the predicted stations deviate from the actual stations. It is performed by searching and measuring distance from predicted location to existing bike station. RMSE measures the dispersion of predictions relative to actual locations using standard formulas. Coverage measures what fraction of actual stations is within a radius R of at least one predicted station. It could be then said that it evaluates how well the predicted network represents the real network.

In this table it can be noticed that average distance to the closest existing station is about 250 [m] and it is lower (better) for training area (Warsaw) than for test area (Łódź), which is non-surprising. The value of RMSE is close to 300 [m] in both cases. It is also important that in the distance up to 300 [m] from predicted stations, there is about 60% of existing locations and this factor increases to about 70% and about 80–90% with increasing coverage distance to 500 [m] and 700 [m]. Especially in the last case, the result of the measure is satisfying. Value of objective function lowers with the increase of coverage distance, which is connected to the coverage factor. It should be mentioned that the lower the function is, the better the result of prediction are.

Calculations described above were also parametrized by changing the value of a threshold of the distance between existing and predicted station. If the prediction is further to the nearest existing station than the value of a threshold, then it is marked as an excessive error and it is not taken into consideration in computations. The values of thresholds used were defined minding the fact that the largest distance between two bike stations both in Warsaw and in Łódź equals about 3000 [m]. Lower values (1000 [m] and 2000 [m]) were defined to show the change distribution in equal intervals. Results depending on the values of aforementioned threshold and with the value of coverage radius of the best result of objective function (700 [m]) are shown in S13 Table.

It is obvious that with the increase of the value of threshold, value of MAE and, by extension, value of RMSE also increase as larger group of predictions located further away from existing locations is taken into account. For the same reason, also the value of objective function increases as it is dependent also on MAE and RMSE. This gives worse results as the lower the function is, the better the result of prediction are, like it was said earlier before.

It is worth mentioning that the use of coverage radius and distance threshold was necessary to define the formal objective function of results, which is one of main indicators of similarity of predicted bike-sharing network to existing ones. The value of this function depends on different values of aforementioned quantities, which shows that the quality of the model is relative and can differ depending on the tolerance of needs of the bike-sharing system.

## Model complexity and computational efficiency

In order to check the quality of tested models of prediction in one more way, a model complexity and computational efficiency of them were examined. The results are shown in S14 Table below.

The content of the S14 Table indicates that the model complexity depends on whether the decision tree with classification, linear regression or ANN was used. As the types of quantities measured differ, they cannot be compared and constitute informational value. Only the results of complexity measurement of ANNs show that the linear regression of results gives more complex models than classification. What definitely can be measured is computational efficiency of models. Nevertheless, it should be mentioned that while decision tree performs only one model building iteration, ANNs do many iterations (equal to the number of epochs applied), so the differences in training and inference time between above mentioned methods can be misleading (time for decision tree methods should be multiplied by the number of epochs used for ANNs to make it more authoritative, which is 200 in this case). Therefore, the best source of comparison should be CPU and RAM usage. On the basis of that it could be said that the Case 5 performs the best results – it has the lowest CPU usage and second lowest RAM usage although having also the most advanced complexity. In addition, the training and inference times are the lowest of all ANNs examined. Reminding that the Case 5 had also the best results in accuracy, precision, sensitivity and specificity matters, the analysis of model complexity and computational efficiency proves additionally that it is the best of all proposed models.

## Implementation of the model

To demonstrate that the proposed methodology has practical applications, the authors of the research performed data preprocessing and prediction of bike station placement in a city without an urban bike system, namely Rzeszów. Fig 13 shows the results of such a prediction. Based on the analysis of the graphic, it can be concluded that, similar to Łódź and Warsaw, the suggested locations are primarily in the city centre, at intersections, and along major roads. Importantly, the spacing between stations is noticeably related to building density – more compact areas have smaller distances between individual stations, which is a crucial feature supporting the comfort level of urban bike users. Thus, the analysis allows for accelerated and partially automated planning of the bike station network distribution in the city of Rzeszów.

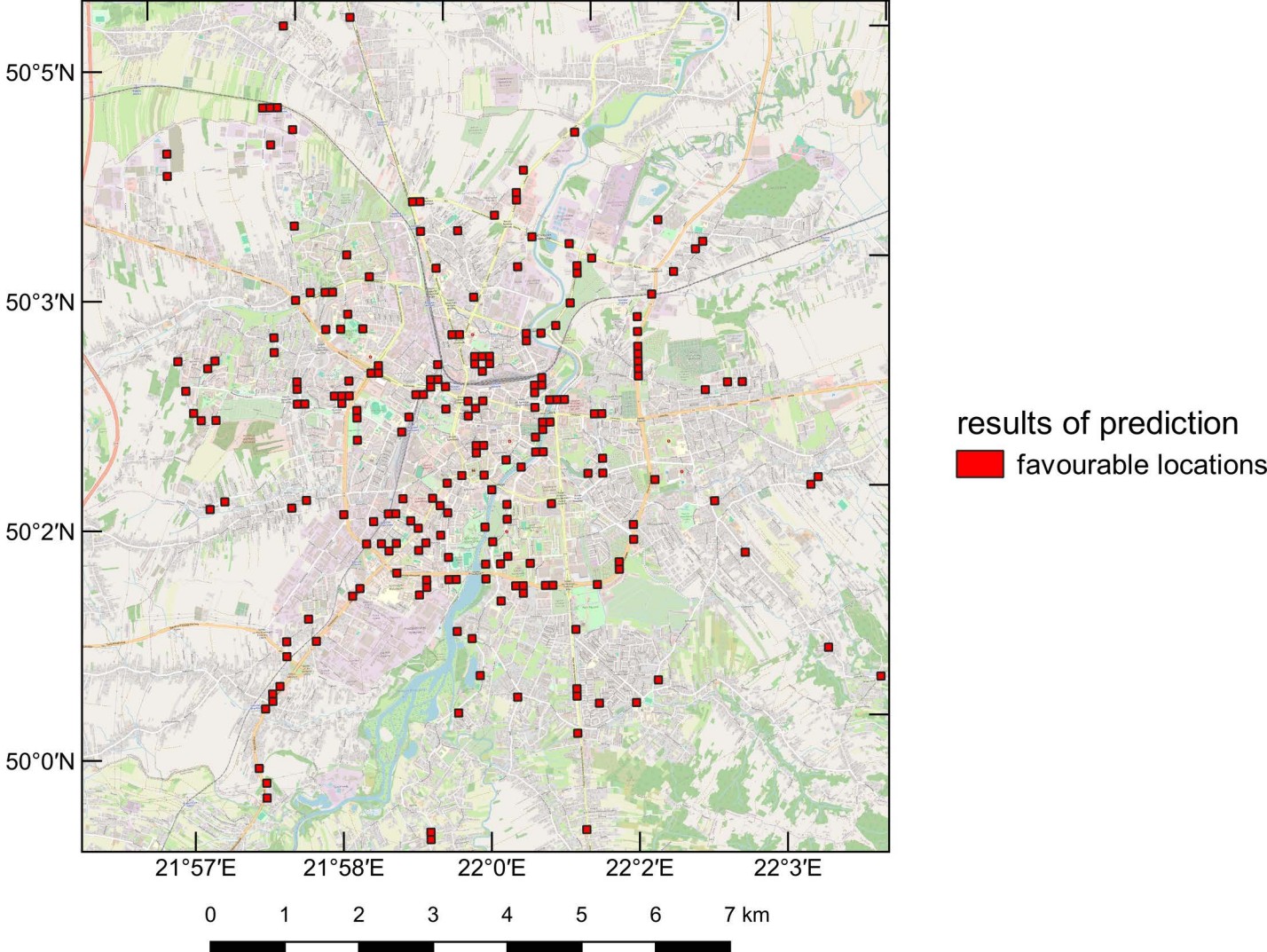

**Fig 13. Prediction results for the bike station network locations in the area of Rzeszów.** The base map created with Quantum Geographic Information System (QGIS) software using OpenStreetMap (https://www.openstreetmap.org/), shared under the Open Database Licence (http://www.opendata-commons.org/licenses/odbl).

## Model use generalizability

It should be noted that although study area and origin of data used are limited to Poland, the applied methodology can be implemented in every city in the world, providing that there are similar types of objects which are part of the urban landscape and that. Moreover, the size of the city should be large enough so that there is a possibility of dividing the terrain of it in at least several hundreds or thousands of primary fields (otherwise the data used in the process of teaching ANN might be not enough). Obviously, the differences in layouts of the cities that come from cultural context can negatively influence the quality of results of algorithm's predictions, yet it has to be mentioned that every city is characterised by its individuality and urban heterogeneity, so the same model would work differently according to the city of its implementation. Also, the differences in data structure and definition of objects used in the study and available to use for different locations worldwide are another threat to the stability of proposed algorithm. Therefore, although the methodology is universal, the created model local. To make it universal, the methodology should be implemented in many cities all over the world. It also will be in area of interest regarding future works.

## Conclusions

The conducted research demonstrates that developing a universal model for precise prediction of bike station locations, based on topographical and demographic data, is feasible. The best results were achieved using a multilayer perceptron (MLP) with a 16-8-4-1 structure and a hyperbolic tangent activation function, which achieved a prediction accuracy of approximately 0.977.

The model was trained on data from Warsaw, and its effectiveness was validated using data from Łódź, where the locations predicted by the model were compared with the actual bike station locations. These results confirmed a high level of coincidence – the model achieved nearly identical prediction accuracy (0.981), and many of the suggested locations indeed matched existing stations. Moreover, the analysed locations were often found near major intersections and public transportation stops, confirming their suitability for the urban bike network. A noticeably higher density of stations was observed in the city centre compared to the outskirts. Similar results obtained from implementation of the method in both aforementioned cities shows that it is possible to use it regardless of the location.

Analysis of the impact of individual variables on the placement of bike stations revealed that the distance from the city centre and proximity to public transportation stops play a key role. Both factors showed a strong correlation with existing locations. Other important variables included distance from roads, population density, and proximity to educational institutions such as schools and universities.

The model also found practical application in the city of Rzeszów, which does not yet have its own urban bike rental system. The analysis results for this city indicated locations similar to those observed in Warsaw and Łódź – near major intersections, transportation hubs, areas with high population density, and educational institutions. This suggests that the model has the potential to support the planning process for bike station placement in new locations.

In summary, the results obtained demonstrate that the developed model can significantly improve the planning and development of bike networks in cities, especially where such systems are still in development. Enhancing the model with additional variables, such as pedestrian and cyclist traffic analysis, could further increase its effectiveness and accuracy. Future research will focus on integrating these factors, allowing the model to be applied not only in Polish cities but also in metropolises worldwide.

## Supporting information

**S1 Table. Supporting and non-supporting elements for bike station location.**
(DOCX)

**S2 Table. Confusion matrix for Case 1.**
(DOCX)

**S3 Table. Confusion matrix for Case 2.**
(DOCX)

**S4 Table. Confusion matrix for Case 3.**
(DOCX)

**S5 Table. Confusion matrix for Case 4.**
(DOCX)

**S6 Table. Confusion matrix for Case 5.**
(DOCX)

**S7 Table. Statistics of tested prediction methods.**
(DOCX)

**S8 Table. Top five factors influencing prediction results.**
(DOCX)

**S9 Table. Percentage of predicted bike station locations within a given distance range from the nearest existing bike station for Warsaw and Łódź.**
(DOCX)

**S10 Table. GMI, p-value, and z-score values for bike station predictions in the areas of Warsaw, Łódź, and Rzeszów.**
(DOCX)

**S11 Table. Accuracy, AUC and MSE as metrics of cross-validation of bike station predictions for Warsaw and Łódź.**
(DOCX)

**S12 Table. MAE, RMSE, Coverage and total value of objective function for predicted and actual locations of bike stations depending on coverage radius.**
(DOCX)

**S13 Table. MAE, RMSE, Coverage and total value of objective function for predicted and actual locations of bike stations depending on threshold value.**
(DOCX)

**S14 Table. Model complexity and computational efficiency of tested prediction methods.**
(DOCX)

## Acknowledgments

A publication fee is funded by Military University of Technology, Faculty of Civil Engineering and Geodesy, Grant Number 531-000130-W400-22. The funders had no role in study design, data collection and analysis, decision to publish, or preparation of the manuscript. We would like to thank the reviewers for their valuable suggestions that led to an improvement of the paper.

## Author contributions

**Conceptualization:** Marek Weis.

**Data curation:** Marek Weis.

**Formal analysis:** Wojciech Dawid.

**Methodology:** Marek Weis, Wojciech Dawid.

**Project administration:** Wojciech Dawid.

**Software:** Marek Weis.

**Supervision:** Wojciech Dawid.

**Validation:** Marek Weis.

**Visualization:** Marek Weis.

**Writing – original draft:** Marek Weis.

**Writing – review & editing:** Wojciech Dawid.

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
