## [Decision Letter · Decision Letter 0]

4 Mar 2026

PONE-D-25-59280Optimizing bike-sharing station locations: a machine learning and artificial neural networks approach using geospatial and demographic dataPLOS One

Dear Dr. Weis,

Thank you for submitting your manuscript to PLOS ONE. After careful consideration, we feel that it has merit but does not fully meet PLOS ONE’s publication criteria as it currently stands. Therefore, we invite you to submit a revised version of the manuscript that addresses the points raised during the review process.

We look forward to receiving your revised manuscript.

Kind regards,

Muhammad Athar, PhD

Academic Editor

PLOS One

Journal Requirements:

3. Please note that PLOS One has specific guidelines on code sharing for submissions in which author-generated code underpins the findings in the manuscript. In these cases, we expect all author-generated code to be made available without restrictions upon publication of the work. Please review our guidelines at https://journals.plos.org/plosone/s/materials-and-software-sharing#loc-sharing-code and ensure that your code is shared in a way that follows best practice and facilitates reproducibility and reuse.

4. Thank you for stating the following financial disclosure: [A publication fee will be funded by Military University of Technology, Faculty of Civil Engineering and Geodesy].

Please state what role the funders took in the study.  If the funders had no role, please state: " "The funders had no role in study design, data collection and analysis, decision to publish, or preparation of the manuscript." "

5. Thank you for uploading your study's underlying data set. Unfortunately, the repository you have noted in your Data Availability statement does not qualify as an acceptable data repository according to PLOS's standards.

7. We note that Figures 1,2,9,10,11,12 and 13 in your submission contain map images which may be copyrighted. All PLOS content is published under the Creative Commons Attribution License (CC BY 4.0), which means that the manuscript, images, and Supporting Information files will be freely available online, and any third party is permitted to access, download, copy, distribute, and use these materials in any way, even commercially, with proper attribution. For these reasons, we cannot publish previously copyrighted maps or satellite images created using proprietary data, such as Google software (Google Maps, Street View, and Earth). For more information, see our copyright guidelines: http://journals.plos.org/plosone/s/licenses-and-copyright.

1. You may seek permission from the original copyright holder of Figures 1,2,9,10,11,12 and 13 to publish the content specifically under the CC BY 4.0 license.

Please upload the completed Content Permission Form or other proof of granted permissions as an " "Other" " file with your submission.

9. We notice that your supplementary file tables are included in the manuscript file. Please remove them and upload them with the file type 'Supporting Information'. Please ensure that each Supporting Information file has a legend listed in the manuscript after the references list.

Reviewers' comments:

Reviewer's Responses to Questions

**Comments to the Author**

1. Is the manuscript technically sound, and do the data support the conclusions?

Reviewer #1: Yes

Reviewer #2: Yes

2. Has the statistical analysis been performed appropriately and rigorously? 

Reviewer #1: Yes

Reviewer #2: Yes

3. Have the authors made all data underlying the findings in their manuscript fully available?

Reviewer #1: Yes

Reviewer #2: Yes

4. Is the manuscript presented in an intelligible fashion and written in standard English?

Reviewer #1: Yes

Reviewer #2: Yes

5. Review Comments to the Author

Reviewer #1: This paper presents a novel machine learning approach for optimizing bike-sharing station locations using geospatial (BDOT10k) and demographic (GHS-POP) data. The authors develop a predictive model using decision trees, random forests, and artificial neural networks (ANN) to identify optimal station locations. The paper is well-structered but should be revised before publication.

1. The paper uses a 100x100m grid due to GHS-POP data constraints but does not justify this choice. It is recommended that authors should conduct and report comparative experiments with alternative grid sizes to validate the selected grid size.

2. While multiple models are tested, the paper does not sufficiently explain why ANN with tanh activation outperformed other models. Add a comparative analysis of model complexity, computational efficiency, and interpretability to clarify the model selection rationale.

3. The finding that "bicycle lanes" is not a significant factor contradicts common expectations.Please provide deeper discussion of this finding, potentially linking it to the underdeveloped bicycle infrastructure in Polish cities.

4. The study is limited to Polish cities without exploring applicability in different cultural or urban contexts.

5. Provide specific information on the over-sampling technique used (e.g., SMOTE parameters) and its impact on model performance.

Reviewer #2: The manuscript addresses an important and practical problem in urban transportation planning by proposing a machine learning and artificial neural network–based framework for optimizing bike-sharing station locations. Nevertheless, several methodological, analytical, and presentation issues should be carefully addressed to further strengthen the scientific rigor and clarity of the paper.

(1) While the manuscript combines machine learning, artificial neural networks, and geospatial analysis in a coherent framework, the methodological innovation relative to existing studies is not sufficiently articulated.

(2) The selection of supportive and non-supportive spatial factors (Table 1) appears reasonable; however, the rationale behind including or excluding specific variables remains largely qualitative. A more systematic justification would improve the transparency and reproducibility of the proposed methodology.

(3) The manuscript emphasizes that the model is “applied in metropolises worldwide” ,as the model is trained mainly on Warsaw data and only validated in Łódź and Rzeszów. Limitations related to urban heterogeneity and model generalizability should be discussed.

(4) The results show high sensitivity but low specificity, suggesting a class imbalance. The manuscript should clarify whether any imbalance-handling techniques were applied.

(5) The analytical roles of coverage radius and distance threshold (Tables 12 and 13) are not clearly distinguished, and the chosen threshold values (1000–3000 m) lack behavioral or planning-based justification.

(6) Rather than simply enumerating studies, the literature review should synthesize prior findings into overarching insights. At present, the reviewed literature does not sufficiently support a conclusion such as “Based on the reviewed literature, it is clear that the success of bike-sharing programs heavily depends on the strategic placement of bike stations,”

(7) The manuscript proposes a two-stage prediction framework, in which decision tree or random forest methods are applied prior to artificial neural networks (ANN). However, the rationale behind adopting this two-stage structure, as well as its advantages over a single-model approach, is not clearly explained and should be further elaborated.

(8) According to the journal’s reference formatting guidelines, references should be numbered based on the order of their first appearance in the text rather than being sorted alphabetically. In addition, the resolution of some figures could be further improved; and the first-line indentation is inconsistent across several paragraphs. A careful formatting revision is recommended.

6. PLOS authors have the option to publish the peer review history of their article (what does this mean?). If published, this will include your full peer review and any attached files.

Reviewer #1: **Yes:** Gen Li

Reviewer #2: No

---

## [Author Response · Author response to Decision Letter 1]

30 Mar 2026

Dear Editor and Reviewers,

We would like to thank you for spending your precious time on processing the manuscript and for giving us constructive suggestions to deeply improve the quality of the paper. Please find the detailed responses in "Response to Editor" and "Response to Reviewers" files and the corresponding revisions in track changes in the re-submitted files. We hope that our answers are satisfactory and in line with your comments. All comments have been answered and either implemented in the work or explained why no action was taken. Thanks to them the substantive value of the article was improved and we hope that this article will be published.

Kind regards,

The Authors

---

## [Decision Letter · Decision Letter 1]

28 Apr 2026

Optimizing bike-sharing station locations: a machine learning and artificial neural networks approach using geospatial and demographic data

PONE-D-25-59280R1

Dear Dr. Weis,

We’re pleased to inform you that your manuscript has been judged scientifically suitable for publication and will be formally accepted for publication once it meets all outstanding technical requirements.

Kind regards,

Muhammad Athar, PhD

Academic Editor

PLOS One

Additional Editor Comments (optional):

Reviewers' comments:

Reviewer's Responses to Questions

**Comments to the Author**

1. If the authors have adequately addressed your comments raised in a previous round of review and you feel that this manuscript is now acceptable for publication, you may indicate that here to bypass the “Comments to the Author” section, enter your conflict of interest statement in the “Confidential to Editor” section, and submit your "Accept" recommendation.

Reviewer #1: All comments have been addressed

Reviewer #2: All comments have been addressed

2. Is the manuscript technically sound, and do the data support the conclusions?

Reviewer #1: Yes

Reviewer #2: Yes

3. Has the statistical analysis been performed appropriately and rigorously? 

Reviewer #1: Yes

Reviewer #2: Yes

4. Have the authors made all data underlying the findings in their manuscript fully available?

Reviewer #1: Yes

Reviewer #2: Yes

5. Is the manuscript presented in an intelligible fashion and written in standard English?

Reviewer #1: Yes

Reviewer #2: Yes

6. Review Comments to the Author

Reviewer #1: The authors have addressed all my comments and the manuscript has been improved. I the it can be accepted now.

Reviewer #2: (No Response)

7. PLOS authors have the option to publish the peer review history of their article (what does this mean?). If published, this will include your full peer review and any attached files.

Reviewer #1: **Yes:** Gen Li

Reviewer #2: No

---

## [Editor Report · Acceptance letter]

PONE-D-25-59280R1

PLOS One

Dear Dr. Weis,

I'm pleased to inform you that your manuscript has been deemed suitable for publication in PLOS One. Congratulations! Your manuscript is now being handed over to our production team.

Kind regards,

on behalf of

Dr. Muhammad Athar

Academic Editor

PLOS One